# Impact of Thermal Pulsation System Therapy on Pre-Operative Intraocular Lens Calculations before Cataract Surgery in Patients with Meibomian Gland Disfunction

**DOI:** 10.3390/medicina59040658

**Published:** 2023-03-27

**Authors:** Paulina Szabelska, Joanna Gołębiewska, Radosław Różycki

**Affiliations:** Department of Ophthalmology, Military Institute of Aviation Medicine, 01-755 Warsaw, Poland

**Keywords:** cataract surgery, intraocular lens power, IOL calculation, cataract, MGD, Meibomian gland dysfunction, Thermal Pulsation System, TPS

## Abstract

In patients with eye surface disorders such as dry eye syndrome or Meibomian gland dysfunction (MGD) it is necessary to improve the tear film condition in order to obtain visual system measurements before cataract surgery. The aim of the project was to analyze the Thermal Pulsation System (TPS) impact on the visual system parameters used in cataract surgery qualification. The study included six patients (11 eyes) with MGD diagnosis. All patients were treated with TPS. The obtained results were compared and used to calculate the power and type of the intraocular lens (IOL). As a treatment result, the power of astigmatism has changed in 64% of the eyes. Planned surgical treatment type has changed in 27% of cases. TPS also affected the cylinder axis in three eyes, which was 27% of cases. Based on the calculations, power of the recommended IOL has changed in five eyes (46%). Stabilization of visual system parameters after TPS allowed to improve the accuracy of the results. It also ensured the proper astigmatism treating method during cataract surgery and allowed selection of the proper IOL power and type.

## 1. Introduction

The latest surgical technologies made it possible to combine cataract surgery with other vision defects removal, e.g., astigmatism or presbyopia. Modern intraocular lens (IOL) selection depends on the patient’s needs and expectations, as well as local conditions prevailing in the eye. The basic IOL types include monofocal, multifocal and toric lenses [1]. Contemporary IOLs are made of good quality material and equipped with appropriate filters. Because of that, they not only restore patients’ good visual acuity and correct concomitant vision defects, but also protect eyes after cataract surgery [1,2].

Dry eye syndrome (DES) can be caused by a lipid layer of the tear film disorder and associated with obstructed Meibomian glands located in the eyelids. This condition is known as Meibomian Gland Dysfunction (MGD). Typical symptoms presented by patients with MGD include foreign body sensation, burning, dryness, chronic redness of the eyelid edge and conjunctiva or visual disturbances [3]. MGD affects more than 340 million people worldwide. Clinical studies indicated that more than 60% of cataract patients with DES did not complain of its symptoms before surgery [4]. In addition, patients who did not report DES signs before surgical procedure can report them after treatment. Therefore, DES and MGD diagnostics should be routine in people referred for surgical intervention on the eye surface, which is recommended by the American Society of Cataract and Refractive Surgery (ASCRS) guidelines [5].

In MGD treatment, e.g., warm compresses and a self-made eyelid massage, can be used [6]. However, during massage, patients are not able to control the pressure on the eyelids, its impact on the surface area and duration.

A similar treatment effect can be achieved by using moisturizing eye drops before and after the surgical procedure. They allow tear film layer supplementing. Unfortunately, this method does not remove the MGD cause; it is only a symptomatic treatment. The eye drop effectiveness largely depends on the patient’s cooperation; a patient may forget about eye drop application. For this reason, Thermal Pulsation System (TPS) is a good solution in MGD therapy. This method combines mechanical unblocking of the Meibomian glands and heating the secretion produced by them. The device uses a constant, safe temperature throughout the massage [7,8,9].

The aim of the study was to determine the ways in which TPS affects the parameters of the visual system and the results of tests performed before cataract surgery with special focus on IOL calculation. The study also indicated TPS impact on IOL type selection, as well as on the expected postoperative astigmatism (including setting of its axis).

## 2. Materials and Methods

This is a case series of patients with a diagnosis of MGD, DES and blepharitis, confirmed by clinical examination. The patients attended our outpatient clinic from May 2022 until August 2022. In total, 60 patients were examined. We detected symptoms of three disorders—blepharitis, MGD and DES in 6 patients who were subsequently included in the study (11 eyes; 1 eye was excluded due to absence of all diseases signs). All of them were Caucasian between 42 and 82 years old (mean age was 67 y.o.). The examination was performed before TPS treatment and 6 weeks (+/−1 week) after TPS. The same parameters were checked during appointments. Patients were complaining about DES symptoms such as burning, redness of the eyes and foreign body sensation for minimum 3 months or more. They were carefully interviewed before planned cataract surgery. Blepharitis, MGD and DES were confirmed in slit-lamp examination. Patients who qualified for this study had moderate signs of diseases. They had one or two folds in nasal and temporal part of conjunctiva. All the individuals were examined with Efron Grading Scale (EGS) which provided a convenient clinical reference. All the patients who qualified for the study had stage 1 or 2 EGS—pale or red lid margin, opening of Meibomian glands were visible, with or without yellow crust at base of lashes and some of the lashed were stuck together. In addition, conjunctival redness was increased in all patients—stage 1 or 2 in EGS scale. All the individuals were diagnosed with 1 or 2 stage of MGD; they had cloudy or milky expression at most gland orifices and had or did not have increased tearing.

Photographs were taken before and after TPS treatment (Figure 1).

There was no history of ocular surgeries and topical medication use in all participants. DES symptoms were also assessed with the Ocular Surface Disease Index (OSDI), Schirmer test and tear breakup time (TBUT) before and after performed treatment.

Diagnostic tests included measurement of intraocular pressure (IOP) using the Canon TX-20P air-puff tonometer and visual acuity (VA) using Snellen charts. Subsequently, autorefractometry and keratometry were performed in all patients using the Topcon KR-800. The best corrected visual acuity (BCVA) was checked. At the end, biometrics (IOL Master 500) was also performed. The SRK-T formula was used for all IOL calculations. All measurements were conducted for the same IOL type. Measurements were performed 5 times. In this study, the average results are presented. After pre-operative visit, all patients underwent TPS treatment. The procedure is shown on Figure 2.

Our study was in line with the provisions of the Declaration of Helsinki on human studies, and from all those subjects, informed consent to participate in this study was obtained.

Study data were analyzed with the Statistical Package for Social Sciences (SPSS) for Windows version 22.0 (SPSS Inc., Chicago, IL, USA).

## 3. Results

The case series consisted of six patients (11 eyes). Each eye included in the study was marked with the numbers 1–11. Improvement of ocular surface stability and MGD with concomitant DES symptoms was presented in clinical examination and shown in Figure 1 in Materials and methods section.

Schirmer test, TBUT and OSDI results are presented in Table 1.

The mean ($\bar{x}$) Schirmer test value before therapy was 10.27, with a median (Me) of 10.00, and after treatment $\bar{x}$ = 13.09 with Me = 13.00. In addition, results of TBUT test before and after treatment has changed (before TPS $\bar{x}$ = 9.09, Me= 9.00 and after TPS $\bar{x}$ = 13.91 and Me = 14.00). OSDI values have showed a decrease in patient complaints (pre-TPS $\bar{x}$ = 31.09, Me = 35.00 and post-TPS $\bar{x}$ = 22.73 and Me = 27.00).

All pre-operative measurements were obtained before and after TPS. Table 2 shows the results of keratometry (K1, K2) performed before and after TPS. All values of K1 and K2 were taken as “K total”. The keratometry values were, respectively:
Before TPS: K1 mean ($\bar{x}$) = 42.85, median (Me) = 43.64; K2: $\bar{x}$ = 42.89, Me = 43.81; K total $\bar{x}$ = 43.24, Me = 43.16.After TPS: K1 $\bar{x}$ = 42.99, Me = 43.06; K2: $\bar{x}$ = 43.70, Me = 43.65; K total $\bar{x}$ = 43.39, Me = 43.40.

The results are presented in Table 2 provided in diopters [D].

Figure 3 shows vector graphs of keratometric astigmatism before and after TPS. The mean cylinder power value before therapy was 0.79, with a Me of 0.70, and after treatment $\bar{x}$ = 0.84 with Me = 0.70.

Figure 4 shows the keratometric astigmatism changes after TPS compared to the baseline and are presented as absolute values. The average value by which the power of astigmatism changed was 0.21, with Me at 0.11.

Astigmatism increased or decreased after TPS in 64% of the eyes. As a definition of change in the astigmatism power, its value equal to or greater by 0.1 D was assumed. During analysis of astigmatism for each eye, there was an increase in its power in 36% of the eyes and decrease in 28%. In 36% of the eyes, the power of astigmatism remained unchanged.

TPS also affected the cylinder axis in three eyes: in two, it has changed from oblique (OBL) to against the rule (ATR), and in one eye it changes from with-the-rule (WTR) to OBL (based on 30-degree definition). Final results are presented in Table 3. Values for categories (WTR, ATR, OBL) are provided in degrees. The exact changes for each eye are shown in Table 4. In Table 4, axis values are provided in diopters.

Based on the performed measurements, the choice of the recommended IOL power changed in five eyes, which was 46% of the study group. In four eyes, the power value switched downwards, in one eye it changed upwards, which is presented in Figure 5. Change in the IOL power did not exceed 0.5 D in each case.

## 4. Discussion

The effectiveness of DES treatment with eye drops largely depends on the patient’s compliance; a patient may forget about eye drop application. For this reason, TPS is a good solution in MGD treating. This method combines mechanical unblocking of the Meibomian glands and heating the secretion produced by them. The device uses a constant, safe temperature throughout the duration of the massage [6,7,8,9]. Due to the measurable parameters of the performed massage, i.e., constant pressure on the eyelids, constant temperature of 42.5 degrees Celsius and protection of the eye surface against the side effects of excessively high temperature, the TPS system seems to be a good and safe tool for the MGD treatment. Unfortunately, the high procedure prices do not allow every medical center to treat patients by TPS.

The main task of a doctor performing visual system measurements and ophthalmological examinations is the proper qualification for the surgery. The calculation of artificial IOL should meet the postoperative patient’s expectations and guarantee a satisfactory quality of life, as well as an acceptable result of the performed procedure. It is also important that the predictions about the implanted IOL power and its type coincide with the previously predicted results of postoperative examinations.

The conducted observations concerned mainly disfunction of the meibomian glands in patients qualified for cataract surgery, as well as the presence of clinical features in the ophthalmological examination and deviations in the test results [5,10,11]. In one publication, 68% of patients who underwent surgery had a change in astigmatism power (52% of patients experienced an increase, 24% decreased) [12]. Our study also indicated that the power of astigmatism after TPS can change in more than 60% of the eyes. Matossian C. in her publication noted changes in the cylinder axis in 7 out of 25 eyes included into the study, which was 28% of all cases. This also coincides with the data obtained in our project.

Stabilization of the tear film after TPS may also affect the decision of using a toric lens or making relaxation cuts to correct astigmatism in almost 30% of patients [12]. We obtained similar results in our study. The planned astigmatism treatment changed in three eyes (27% of all cases). In one patient, we planned a spherical IOL implantation with a relaxation corneal incision. After TPS, we decided to change procedure to toric IOL implantation without making an incision. In two cases, we recommended making peripheral relaxation incisions, which was not planned before TPS.

Looking for a possible change in the power of astigmatism (increase or decrease) and its axis, we conducted an analysis of the relation of the astigmatism power change after TPS and compared to its output power (Figure 6A) and axis (Figure 6B). In the comparisons, we did not detect correlations indicating an unambiguous direction of parameters changes that may occur after TPS.

The measurements conducted in our project did not clearly indicate the direction of changes in power and the axis of astigmatism but have shown that such changes occur in patients and these are not just single cases. The results of our study and those carried out so far have confirmed that untreated MGD before cataract surgery may lead to insufficient or excessive correction of the astigmatism power and changes in its axis. In addition, the available literature suggested that the postoperative astigmatism after TPS treatment allows for better results of surgical procedure in about 28% of individuals [12].

Matching IOL power to visual system parameters may also be inaccurate as a result of the untreated eye surface disease. In our study, the decision of choosing the recommended IOL power changed in 46% of all cases (by a value of 0.5 D). Such a difference, taking into account the possible uncorrected astigmatism, can significantly worsen the quality of vision after the surgical procedure. To our knowledge, this is the first study designed to compare recommended IOL power before and after TPS treatment and paying attention on patients’ clinical examination including photographic documentation when the ocular surface condition collides with ocular surgery.

The results have indicated that patients with DES and MGD referred for cataract surgery must be thoroughly examined and treated before measurements of the visual system and deciding about the type and power of the IOL.

Disadvantages of our study include a small group of patients and the fact that the treatment method is quite expensive and not every medical center has access to it. Therefore, due to the fact that our case series included patients with only moderate stage of disease, it is difficult to indicate the ways in which the parameters of the visual system can change in different disorder severities due to TPS treatment.

## 5. Conclusions

Since the life expectancy is constantly increasing, we strive to ensure that patients who have implanted IOL once in their lives enjoy the best possible quality of vision and visual acuity. Due to the beneficial results achieved after TPS treatment, choosing it as the method of MGD treatment seems to be a good solution. TPS results in unblocking of the Meibomian glands, which cause in long-term stabilization of the eye surface and improve the results of the IOL calculation. Therefore, in our opinion, this system should be considered as a therapeutic method of MGD before the selection of IOL and cataract surgery. Treatment of ocular surface diseases is crucial in proper pre-operative patients care, especially in the meaning of accurate IOL calculations. The results of this study have confirmed that untreated MGD before cataract surgery may lead to insufficient or excessive correction of the astigmatism power and changes in its axis.

The results of our study and other projects carried out so far have shown that it is advisable to repeat the measurements on larger patients’ group to accurately assess the benefits of TPS before cataract surgery.

The outcome of our study will allow to broaden the knowledge about the proper selection of IOLs before cataract surgery and will indicate ways to better predict the postoperative acuity and quality of vision. The multifactorial nature of this study may arouse the interest of not only ophthalmic surgeons, but also doctors working in clinics and will make it possible to direct patients to appropriate therapeutic pathways.

## Figures and Tables

**Figure 1 medicina-59-00658-f001:**
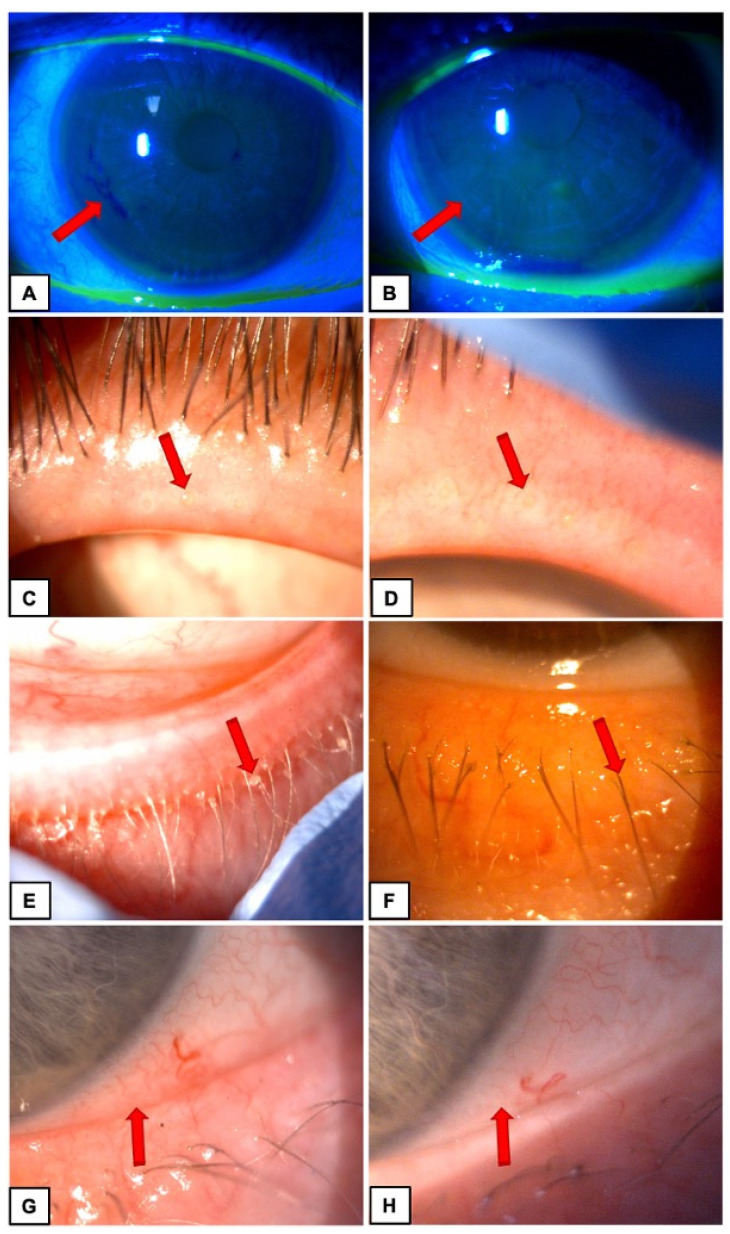
Patients before (**A**,**C**,**E**,**G**) and after (**B**,**D**,**F**,**H**) TPS treatment. Changes between photographs marked by red arrows. Patient 1: TBUT in tenth second of examination (**A**) pre-TPS—brake up of tear film (**B**) post-TPS—no-brake up of tear film. Patient 2: Meibomian gland (**C**) plugged pre-TPS (**D**) permeable post-TPS. Patient 3: Pathological secretion material around eyelashes (**E**) present (**F**) absent. Patient 4: Ocular conjunctival redness (**G**) intermediate—level 2 (**H**) low—level 1.

**Figure 2 medicina-59-00658-f002:**
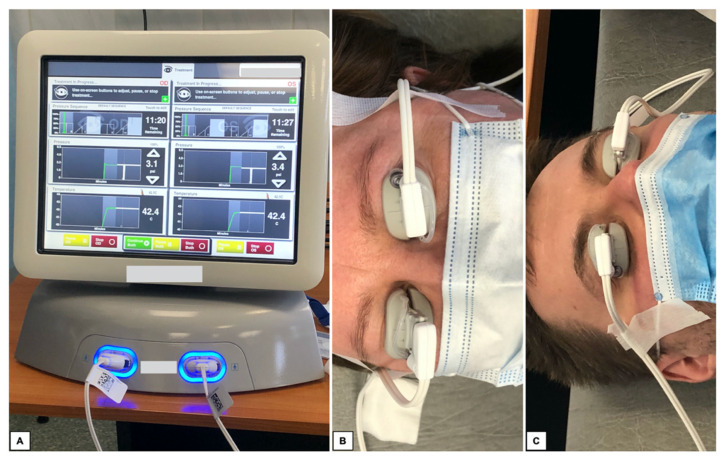
TPS treatment. (**A**) TPS device (**B**,**C**) Patients during TPS session.

**Figure 3 medicina-59-00658-f003:**
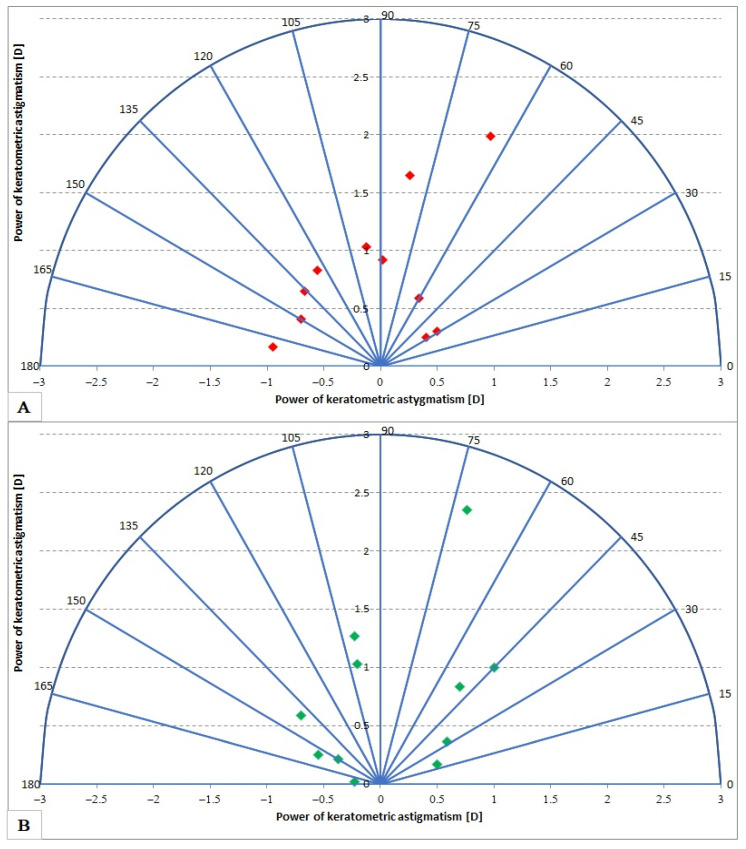
Keratometric astigmatism before (**A**) and after (**B**) TPS.

**Figure 4 medicina-59-00658-f004:**
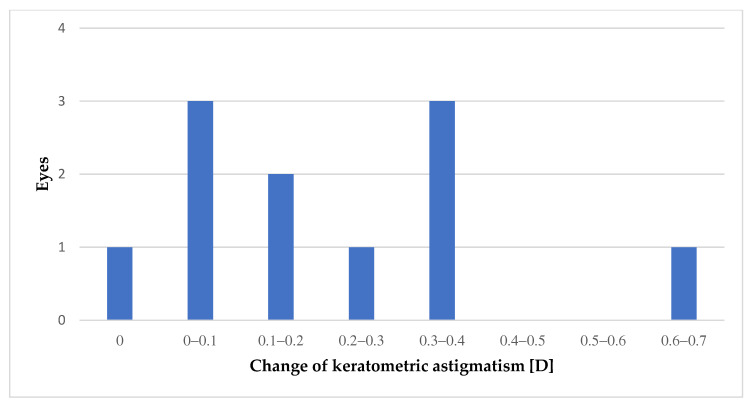
Change in keratometric astigmatism after TPS.

**Figure 5 medicina-59-00658-f005:**
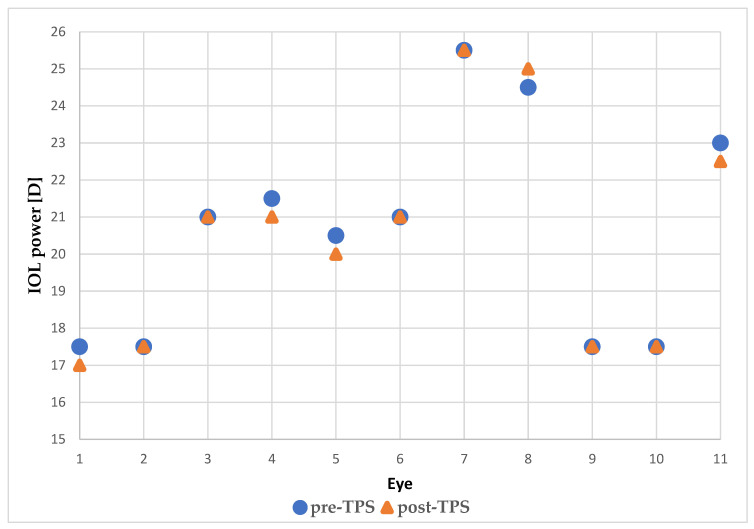
The power of the recommended IOL before and after TPS.

**Figure 6 medicina-59-00658-f006:**
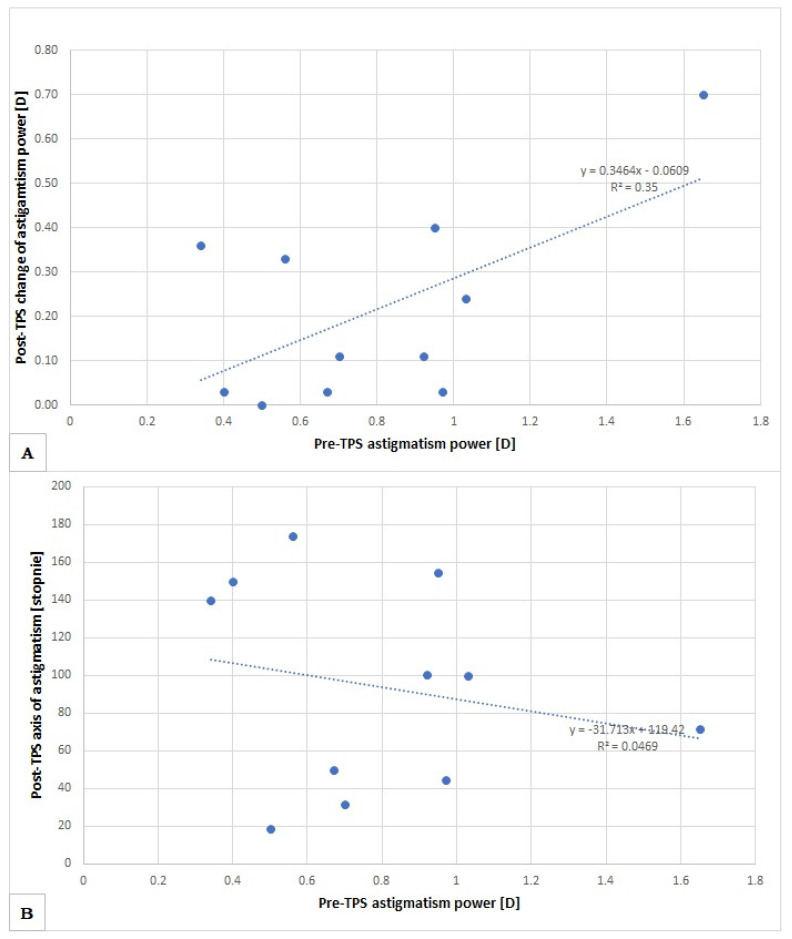
Changes in astigmatism power after TPS compared to its output power (**A**) and axis (**B**).

**Table 1 medicina-59-00658-t001:** Schirmer test, TBUT and OSDI results before and after TPS.

Eye	TPS Treatment Time					
	Pre-TPS			Post-TPS		
	Schirmer Test (mm)	TBUT (s)	OSDI (Points)	Schirmer Test (mm)	TBUT (s)	OSDI (Points)
1	10	10	35	12	13	27
2	11	9	35	14	11	27
3	10	9	37	11	13	30
4	9	8	37	11	12	30
5	10	7	40	13	14	28
6	9	8	40	13	12	28
7	11	9	21	15	17	15
8	12	10	21	15	17	15
9	11	10	25	14	15	17
10	10	9	25	14	15	17
11	10	11	26	12	14	16

**Table 2 medicina-59-00658-t002:** Changes in keratometry values before and after TPS treatment.

Eye	K1			K2		
	Pre-TPS	Post-TPS	Change	Pre-TPS	Post-TPS	Change
1	42.06	42.10	0.04 ↑	42.40	42.80	0.40 ↑
2	42.00	42.05	0.05 ↑	42.40	42.42	0.02 ↑
3	43.10	43.15	0.05 ↑	43.77	43.85	0.08 ↑
4	43.00	43.06	0.06 ↑	43.70	43.65	0.05 ↓
5	43.55	43.50	0.05 ↓	44.58	44.77	0.19 ↑
6	43.40	43.35	0.05 ↓	44.32	44.38	0.06 ↑
7	42.94	43.05	0.11 ↑	43.89	43.60	0.29 ↓
8	43.05	42.99	0.06 ↓	43.55	43.49	0.06 ↓
9	42.99	43.21	0.22 ↑	43.55	43.44	0.11 ↓
10	42.24	42.72	0.48 ↑	43.21	43.72	0.51 ↑
11	42.99	43.44	0.45 ↑	44.64	45.79	1.15 ↑

**Table 3 medicina-59-00658-t003:** The axis of keratometric astigmatism before and after TPS.

	TPS Treatment			
	Pre-TPS K Axis		Post-TPS K Axis	
Ax of Astigmatism	n	%	n	%
WTR (60–120)	4	36%	3	27%
ATR (0–30, 150–180)	1	9%	3	27%
OBL (30–60, 120–150)	6	55%	5	46%
Total	11	100%	11	100%

**Table 4 medicina-59-00658-t004:** Value and degree of astigmatism for each eye before and after TPS.

Eye	Pre-TPS			Post-TPS		
	Axis Value	Axis Degree	Axis Category	Axis Value	Axis Degree	Axis Category
**1**	0.34	60	OBL	0.70	140	OBL
**2**	0.42	32	OBL	0.37	150	OBL
**3**	0.67	136	OBL	0.70	50	OBL
**4**	0.70	150	OBL	0.59	32	OBL
**5**	1.03	97	WTR	1.27	100	WTR
**6**	0.92	89	WTR	1.03	101	WTR
**7**	0.95	170	ATR	0.55	155	ATR
**8**	0.50	31	OBL	0.50	19	ATR
**9**	0.56	124	OBL	0.23	174	ATR
**10**	0.97	64	WTR	1.00	45	OBL
**11**	1.65	81	WTR	2.35	72	WTR

## Data Availability

Data are available from the corresponding author upon reasonable request.

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
