# Peer review of "Impact of Thermal Pulsation System Therapy on Pre-Operative Intraocular Lens Calculations before Cataract Surgery in Patients with Meibomian Gland Disfunction"

_medicina, 2023, doi:10.3390/medicina59040658_

Round 1

Reviewer 1 Report

This pilot study must have details about the time interval of the procedure. The measurements made in this study did not clearly indicate the direction of changes in power and the axis of astigmatism but have shown that such changes occur in patients and these are not just single cases. The results of this study have confirmed that untreated MGD before cataract surgery may lead to insufficient or excessive correction of the astigmatism power and changes in its axis. In addition, the available literature suggested that the postoperative astigmatism after TPS treatment allows for better results of surgical procedure in about 28% of individuals, but the number of patients /eyes/ is not sufficient and the time interval is not clearly defined.

Author Response

Dear Reviewer, 

Thank you for your constructive and informative review of our paper. We have answered each of your points below.

This pilot study must have details about the time interval of the procedure. The measurements made in this study did not clearly indicate the direction of changes in power and the axis of astigmatism but have shown that such changes occur in patients and these are not just single cases. The results of this study have confirmed that untreated MGD before cataract surgery may lead to insufficient or excessive correction of the astigmatism power and changes in its axis. In addition, the available literature suggested that the postoperative astigmatism after TPS treatment allows for better results of surgical procedure in about 28% of individuals, but the number of patients /eyes/ is not sufficient and the time interval is not clearly defined.

We have added time details of the procedure to the Materials and methods section of the manuscript

Patients have attended our outpatient clinic from May 2022 till August 2022. First examination was performed before TPS treatment. A follow-up examination after TPS treatment was performed after 6 weeks (+/- 1 week). In total, 60 patients were examinated. We found symptoms of three disorders - blepharitis, MGD and DES only in 6 patients, who were included to the study. We also inserted Table 4 to clarify changes between category of astigmatism (ATR, WTR, OBL) and marked those eyes in which axis category has changed by red color. We marked increase in values of astigmatism by orange color, decrease by purple color in this Table. What is more manuscript was revised by English native speaker.

Reviewer 2 Report

In this paper, the effect of thermal pulsation system (TPS) therapy on intraocular lens calculations performed prior to cataract surgery in patients with Meibomian gland dysfunction is examined. TPS therapy is found to have an impact on both the type of IOL chosen and the anticipated amount of postoperative astigmatism (including setting of its axis).

1) The author says in the Materials and Methods section that "Patients had reported experiencing DES symptoms. Slit-lamp examination showed blepharitis, MGD, and DES." All three disorders were present in all six patients? If not, did the severity of the difference between the parameters of the visual system before and after TPS therapy depend on the type of disease? Is it possible to add a table with the patients' details in it?

2) The eye's number in Figure 3 is not labeled. "TPS also impacted the cylinder axis in three eyes," the author writes. "In two it moved from oblique (OBL) to against the rule (ATR), and in one from with-the-rule (WTR) to OBL (based on 30 degrees definition)." Yet Figure 3 does not imply that. Is it possible to modify Figure 3 such that it displays this data?

3) According to the author "For each eye, there was an increase in astigmatism power during analysis in 36% of the eyes and a decrease in 28% of the eyes. The degree of astigmatism was unaltered in 36% of the eyes." It is impossible to tell whether the change in astigmatism is a positive or negative number because datas in Figure 4 are given as absolute values. Is Figure 4 changeable?

4) According to the author, "Based on the performed measurements, the recommendation for the recommended IOL power altered in 5 eyes, which represented 46% of the study group. " What degree of change is there? Is this definition of "IOL power change" equivalent to the one found in "As a definition of change in the astigmatism power its value equal to or greater by 0.1 D"? Is it feasible to label Figure 5's precise value?

Author Response

Dear Reviewer, 

Thank you for your constructive and informative review of our paper. We have answered each of your points below.

In this paper, the effect of thermal pulsation system (TPS) therapy on intraocular lens calculations performed prior to cataract surgery in patients with Meibomian gland dysfunction is examined. TPS therapy is found to have an impact on both the type of IOL chosen and the anticipated amount of postoperative astigmatism (including setting of its axis).

  1. The author says in the Materials and Methods section that "Patients had reported experiencing DES symptoms. Slit-lamp examination showed blepharitis, MGD, and DES." All three disorders were present in all six patients? If not, did the severity of the difference between the parameters of the visual system before and after TPS therapy depend on the type of disease? Is it possible to add a table with the patients' details in it?

We found symptoms of three disorders - blepharitis, MGD and DES in all six patients. Patients qualified to this study had moderate symptoms of diseases.

  1. They had one or two folds in nasal and temporal part of conjunctiva.
  2. All the individuals were examinated with Efron Grading Scale (EGS) which provided a convenient clinical reference. All the patients qualified to the study had stage 1 or 2 EGS - pale or red lid margin, opening of Meibomian glands were visible, with or without yellow crust at base of lashes and some of the lashed were stucked together. Also, conjunctival redness was increased in all patients - stage 1 or 2 in EGS scale. All the individuals were diagnosed with 1 or 2 stage of MGD - they had cloudy or milky expression at most gland orifices and had or did not have increased tearing.

Our case series included patients with only moderate stage of disease it is difficult to indicate how the parameters of the visual system can change in different disorder severities due to TPS treatment.

We have added of those informations to the manuscript.

  1. The eye's number in Figure 3 is not labeled. "TPS also impacted the cylinder axis in three eyes," the author writes. "In two it moved from oblique (OBL) to against the rule (ATR), and in one from with-the-rule (WTR) to OBL (based on 30 degrees definition)." Yet Figure 3 does not imply that. Is it possible to modify Figure 3 such that it displays this data?

We have added Table 4 to clarify those changes and marked those eyes in which axis category has changed by red color.

  1. According to the author "For each eye, there was an increase in astigmatism power during analysis in 36% of the eyes and a decrease in 28% of the eyes. The degree of astigmatism was unaltered in 36% of the eyes." It is impossible to tell whether the change in astigmatism is a positive or negative number because datas in Figure 4 are given as absolute values. Is Figure 4 changeable?

We have added those values in new Table 4. Increased in values was marked by orange color, decrease by purple color in this Table.

  1. According to the author, "Based on the performed measurements, the recommendation for the recommended IOL power altered in 5 eyes, which represented 46% of the study group. " What degree of change is there? Is this definition of "IOL power change" equivalent to the one found in "As a definition of change in the astigmatism power its value equal to or greater by 0.1 D"? Is it feasible to label Figure 5's precise value?

Recommended IOL power is not an equivalent to the change of astigmatism. Figure 5 presented recommended IOL power calculated by IOL Master 500 for SRT-KT formula for the same type of IOL. The figure was created based on those measurements and precise values of IOL are given in this figure (y-axis=IOL power). It is not possible to give more accurate values, because measurements are based on the device calculations and we could not change it.

Round 2

Reviewer 1 Report

Please, the group of patients, 6, resl 11 eyes od nôt sufficient  to give duch conclusion, other  študuješ have more than 25 patients. 

Author Response

Dear Reviewer, 

Thank you for your review of our paper. Despite the extension of the response time to 10 days we cannot increase number of participants in this time. The recruitment of patients to the study takes several weeks, the interval between visits is another 6 weeks. This publication is the outcome of a pilot study conducted at the Ophthalmology Clinic in the Military Institute of Aviation Medicine and it is only an introduction to the current running project. The final results will be presented after its completion. 

Best regards,

Paulina Szabelska, MD